# Do Oncologists Recommend the “Pill” of Physical Activity in Their Practice? Answers from the Oncologist and Patients’ Perspectives

**DOI:** 10.3390/cancers16091720

**Published:** 2024-04-28

**Authors:** Aitor Martínez Aguirre-Betolaza, Ander Dobaran Amezua, Fatma Hilal Yagin, Jon Cacicedo, Jurgi Olasagasti-Ibargoien, Arkaitz Castañeda-Babarro

**Affiliations:** 1Department of Physical Activity and Sport Science, Faculty of Education and Sport, University of Deusto, 48007 Bilbao, Spain; a.martinezdeaguirre@deusto.es (A.M.A.-B.); dobaranander@opendeusto.es (A.D.A.); jurgi.olasagasti@deusto.es (J.O.-I.); 2Department of Biostatistics and Medical Informatics, Faculty of Medicine, Inonu University, Malatya 44280, Turkey; hilal.yagin@inonu.edu.tr; 3Department of Radiation Oncology, Cruces University Hospital, BioCruces Health Research Institute, Osakidetza, 48903 Barakaldo, Spain; jon.cacicedofernandezbobadilla@osakidetza.eus

**Keywords:** physical activity recommendations, oncologist, cancer, survey, barriers

## Abstract

**Simple Summary:**

Nowadays, everyone is aware of the health benefits of physical activity (PA). In the case of cancer, the evidence is strong in favour of PA, but in many cases, the message does not reach patients. For this reason, in the present study, we wanted to find out the points of view of oncologists and their respective patients with regard to the prescription of PA and the possible causes for the message not passing through adequately. We observed that 97% of oncologists said that they prescribe PA in their office, while only 62% of their patients said that they have received these guidelines. It was also observed that those patients who claimed to have received recommendations for PA were more active in their daily lives, walking more days per week and more minutes per day. This study is the starting point for finding out where the discrepancies between oncologist–patient communication are.

**Abstract:**

Objectives: The purposes of this current questionnaire-based study were to analyse whether oncologists prescribed PA to their patients in Spain, as well as the type of exercise recommended, the variables that influence whether or not to recommend it and to compare these recommendations with the values reported by their patients. Methods: Two online questionnaires were designed for this study. The first one, filled in by the oncologists (n = 93), contained aspects such as the attitude or barriers to promoting PA. The second was designed for patients with cancer (n = 149), which assessed PA levels and counselling received from oncologists, among other facets. Results: The majority of oncologists (97%) recommend PA during their consultations. Instead, only 62% of patients reported participating in exercise within the last 7 days. Walking was the most common form of exercise, reported by 50% of participants. Patients who received exercise recommendations from their oncologist walked for more days (*p* = 0.004; ES = 0.442) and more minutes per day (*p* = 0.022; ES = 0.410). The barriers most highlighted by patients were lack of time and not knowing how to perform PA. Conclusion: Oncologists and patients seem to be interested and able to participate in PA counselling and programmes. However, there was a discrepancy between what was reported by oncologists and expressed by patients in terms of recommendations for PA and the modality itself.

## 1. Introduction

In the past decades, there has been a global increase in cancer incidence and mortality worldwide, irrespective of the socioeconomic development of the country. Thus, cancer ranks as one of the leading causes of death worldwide [1]. In the case of Spain, with data collected in 2021, malignant tumours accounted for 25.2% of total deaths, making them the second leading cause of death in the country. A total of 279,260 new cases are expected in 2024, similar to the previous year. However, in the future, growth in the absolute number of cases is expected due to two primarily factors: population growth and ageing [1]. Experts being aware of this situation are using different methods to treat cancer, both pharmacological and non-pharmacological treatments, depending on the type and stage of this disease [2]. However, certain pharmacological treatments or a combination of them can negatively affect patients with cancer in disparate ways. It is known that patients can develop anxiety, depression, and fatigue derived from these treatments [3]. Additionally, post-treatment cardiovascular toxicity may be developed, and the body image can be affected in the long term, among other side effects [4,5]. Therefore, in order to treat these side effects and to support the pharmacological treatment itself, there are different non-pharmacological therapies, including regular physical activity (PA) practice [6].

The benefits of PA have been studied in numerous health conditions. The World Health Organization defines PA as any bodily movement produced by skeletal muscles that requires energy expenditure [7]. This fact has led health professionals to develop PA guidelines and prescribe exercise to patients with numerous pathologies such as multiple sclerosis [8], cardiovascular or respiratory diseases [9], or cancer [10], among others. Although the authors stated that in most cases, PA prescriptions must be individualised by reducing the risk factors associated with each of them [8,11], this prescription is often inadequate. In cancer, in particular, year after year, new studies have emerged with more scientific evidence in favour of prescribing specific exercises for patients with cancer [10].

Several studies have concluded that increased PA and exercise reduce the mortality rate among patients and help them ameliorate the side effects of the treatment itself [12,13]. In addition, PA has been shown to positively affect cancer-related fatigue, which is known to affect a high volume of patients, not only during but also after treatment [14]. In this regard, we find different randomised control trials that take into account aspects such as frequency, intensity, time and type of activity that benefit patients in different aspects [15].

Systematic reviews and meta-analyses have explored the effects of PA on outcomes such as mortality, cancer recurrence, and health-related quality of life [16,17,18]. Other studies have shown that resistance training can improve muscle strength, up to 35%, and lean body mass to a greater extent in patients undergoing treatment and survivors than any other type of intervention, both in a short period (6–12 weeks) and in a medium-term intervention (6 months). These interventions involve certain exercises, including the larger muscles of the body, and prior measurements to determine the intensity with which each patient should work [19,20,21]. Furthermore, another study in 2006 suggested that there is a link between increased muscle mass through resistance training and improved quality of life in cancer survivors [20]. Nevertheless, despite all the scientific literature that we can find in favour of PA for patients with cancer, it has been seen that most oncologists recommend PA but do not prescribe the type of exercise that best fits patients with cancer based on the existing literature, among other reasons due to a lack of knowledge about how to refer to clinical PA guidelines or lack of time during medical appointments [22,23]. This study is a novel starting point and fills a gap in the literature, as no study has previously looked at this issue, bringing together the views of both groups simultaneously (i.e., oncologists and their patients).

With all the above mentioned, the primary aim of the present study was to analyse if oncologists prescribed PA to their patients in Spain, as well as the type of PA recommended, exploring the variables influencing whether PA was prescribed or not. The secondary aim was to compare these recommendations with the values reported by their patients.

## 2. Methods

### 2.1. Study Design

Two online surveys derived from guidelines were conducted between October 2022 and December 2022 to assess both PA recommended by oncologists as well as their patients’ perception of these PA recommendations in Spain.

### 2.2. Participants and Procedures

Two different questionnaires were designed by investigators of the University of Deusto, both written in Spanish, combining multiple choice and scale response options. Multiple choice questions had certain possible answers to tick, but if none of them showed their answer, they could tick the answer “other”, and they had a check box where they could give their free opinion. These questionnaires were sent via email and WhatsApp containing a link within a brief description of the research to the Spanish Society of Medical Oncology (SEOM), which was responsible for disseminating it through the oncologists of Spain. All the oncologists reached (n = 93) were asked to disseminate the second questionnaire to their patients during their routine medical consultations. Both questionnaires were planned to be completed within 10 min and had a slightly different perspective and purpose.

On the one hand, the first questionnaire contained 9 items and was designed for the oncologists to fill in. The main aim of this questionnaire was to find out whether the oncologists recommend PA to their patients or not, what type of activity, and, on the contrary, reasons why not to recommend it. Oncologists were also asked about the amount of PA they performed on a daily basis. In this questionnaire, there were qualitative variables such as “gender” and “who would you consider most appropriate to carry out an exercise intervention?” and quantitative variables such as “average time per week practising PA”.

On the other hand, the second questionnaire contained 15 items and was disseminated through oncologists with a link to Google Forms for the patients to fill in. There were two main objectives of this questionnaire. Firstly, to find out the reasons why patients with cancer do or do not practice PA. Secondly, to detect the information/message received from the oncologists regarding the practice of PA and, if so, what kind of activities were recommended to them. We were interested in knowing the amount of PA patients do as part of their daily lives. Therefore, the questions asked were about the amount of time spent being physically active in the last seven days using the Spanish version of the International Physical Activity Questionnaire (IPAQ-SF) in its short form, which has shown high sensitivity in cancer patients [24]. As in the questionnaire for the oncologists, both qualitative and quantitative questions were asked, such as “Has your oncologist recommended that you practice PA?” or “How many minutes did you do vigorous PA such as heavy lifting, digging, aerobics, or fast cycling?”, respectively. Height and weight were also asked in the questionnaire based on the data obtained in the last medical consultation.

All oncologists invited to participate had to be medical doctors specialising in oncology, currently working in Spain and with patients under their care to whom they could send the specific patient questionnaire. As for the patients, the requirements to participate included having been diagnosed with cancer at some point in their lives, being a patient of the oncologists interviewed, being over 18 years old, and living in Spain. Both oncologists and patients were respectively informed prior to starting their questionnaires about the nature and purpose of the study, as well as contact information, and that by completing the questionnaire, they were giving their consent to participate. All responses received were voluntary and anonymous.

### 2.3. Statistical Analysis

Frequency (n) and per cent (%) values were calculated for the qualitative variables, and Pearson’s, Fisher’s exact, and Yates’ corrected chi-square tests were used where appropriate for the comparison of independent qualitative variables. The suitability of quantitative variables to normal distribution [25] was examined by visual (histogram and probability graphs) and analytical (Shapiro–Wilk Test) methods. The assumption of homogeneity of variances was examined with the Levene test. Descriptive statistics were expressed as the median, interquartile range (IQR) for non-normally distributed quantitative variables and mean ± standard deviation for normally distributed quantitative variables. Mann–Whitney U and Kruskal–Wallis H tests were used to compare two or more groups in terms of variables that did not meet the parametric test assumptions. After the Kruskal–Wallis H test results, the Conover test was used for pairwise comparisons of universally significant variables. A one-way ANOVA test was used to compare more than two groups that met the parametric test assumptions. The effect size of results with a significant *p*-value was calculated using Cohen’s d. Interpretation of effect size was assessed following Cohen’s suggested thresholds: Cohen considered d = 0.2 to be a ‘small’ effect size, d = 0.5 a ‘moderate’ effect size, and d = 0.8 a ‘large’ effect [26]. A value of *p*-value < 0.05 was considered statistically significant. Data analyses were performed using IBM SPSS Statistics (Statistical Package for Social Sciences) for Windows 26.0.

## 3. Results

A total of 149 patients (51.4 ± 10.7 years), 126 (84.6%) female and 23 (15.4%) males, participated in the study. The medians of the patient’s height and weight were 165 cm (IQR = 9) and 65 kg (IQR = 16), respectively. While 74 (49.7%) of the patients were doing PA, 75 (50.3%) were not. A total of 51 (68%) patients were not doing PA for other reasons, 9 (12%) patients were not doing PA due to a lack of knowledge on how to perform PA correctly, and 8 (10.7%) patients were disabled. Among the other causes mentioned by patients, the most repeated were lack of time (49%), tiredness (11.7%), or lack of mood or laziness (11.7%).

Eighty-nine patients (62.2%) reported receiving PA recommendations from their oncologists, and the most recommended physical activities by the oncologist were walking (n = 47, 52.8%) and stretching/mobility activities (n = 15, 16.9%). More than 60% of patients did not reach the minimum PA level recommended by the WHO 2020 [1] (engage in moderate aerobic physical activities for at least 150 to 300 min per week or intense aerobic physical activities for at least 75 to 150 min; or an equivalent combination of moderate and intense activities throughout the week). According to the results of the IPAQ survey, the patients showed more days per week [2 (IQR = 4)] of moderate PA (such as carrying light objects, cycling at a regular pace, or playing double tennis), while fewer days [1.5 (IQR = 3)] were doing strenuous physical activities (such as heavy lifting, digging, aerobics, or fast cycling) (Table 1).

When Table 2 was examined, it was observed that patients who were recommended PA by their oncologists walked for more days (*p* = 0.004; ES = 0.442) and more minutes per day (*p* = 0.022; ES = 0.410). However, in terms of other results related to PA status in Table 3, oncologists’ recommendation for PA was not statistically significant (*p* > 0.05).

From the total sample of 97 oncologists participating in the study, 50 (53.7%) were between 30 and 50 years old, 66 (71.0%) were female, and 27 (29.0%) were male. Ninety oncologists (96.8%) recommended PA to cancer patients, and the most recommended activity was walking (n = 46, 51.1%) (Table 3).

There was no significant difference between age ranges in terms of recommending PA to patients, reasons for not recommending PA, and the person performing the exercise (*p* > 0.05). There was a significant difference in age ranges in terms of the type of PA recommended by the oncologist (*p* < 0.05). Younger oncologists (30–40 years old) recommended more walking compared to older oncologists, while middle-aged oncologists (41–50 years old) recommended more Strength training and stretching/mobility activities or exercises (Table 4).

After reviewing gender differences, there were no significant differences between male and female oncologists in recommending PA to their patients, the recommended type of PA, the reasons for not recommending PA, and the most appropriate person to perform PA (*p* > 0.05).

## 4. Discussion

This survey-based study was conducted with the main objective of discovering the type of PA prescribed by oncologists to their patients and comparing it with the PA recommendations received by these patients. Almost all oncologists (97%) recommended PA to their patients; in other words, only three of the oncologists who participated in the study did not recommend PA of any kind to their patients. This finding was higher than that of the previous literature, which reports approximately 50% of oncologists recommend PA to their patients [27,28,29]. However, according to our data, 37.8% of patients claim not to have received any recommendation for PA from their oncologist, which was not entirely consistent with their oncologists’ reports. This was probably because oncologists do not spend the necessary time or do not give the necessary importance to that recommendation, and it goes unnoticed, or because patients are not able to assimilate all the information given to them in consultations and do not grasp this message. Despite this, it is noteworthy that more than half of the participants in this survey received more PA recommendations from their oncologists, which is consistent with the results found by Pilotto et al. [30], although higher compared to 25% of cancer survivors surveyed in an American study between 2013 and 2015 [31]. Nevertheless, there were three main factors to take into account in the U.S. study and when comparing the differences obtained in our study with respect to the existing literature. Firstly, PA may not be recommended by oncologists but by other health care providers. Secondly, it should be mentioned that this publication is almost 10 years old, and nowadays, there is much more awareness and scientific evidence in favour of prescribing physical activity, although it is still not enough. Thirdly, they also reported that the age of respondents ranged from 65 to 99, with the youngest respondents receiving the most recommendations for PA. Thus, it is possible that there was a tendency for oncologists to recommend PA to patients who are younger or able to exercise regularly without major associated risks.

Although the percentage of oncologists participating in the study who do not recommend PA was minimal, 3.23%, it has been found that the reasons for not doing so were similar to those analysed in other studies carried out on healthcare professionals, with the main reasons being lack of time and not knowing how to recommend PA [30,32].

Over the past few decades, it has been observed that PA plays an important role in cancer treatment. PA has been increasingly used to support traditional therapies to the extent that regular PA is the tool most often recommended and recognised by oncologists [33]. Thus, both oncologists and patients gave a score of 8.2 on a scale of 10 for the importance of PA in patients with cancer. At the same time, they stress the importance of informing the patient at all times of the plan to be followed [15,34]. Accordingly, as in a study carried out on colorectal cancer survivors where the fear option was only chosen by 2 out of the 479 participants [35], in our study, fear was the least chosen reason, probably because patients were aware of the safety of PA. On the other hand, we observed similarities in the main barriers reported by patients, one being a non-treatment-related factor (lack of time) and the other being educational (not knowing how to do PA appropriately) [36,37].

When addressing the question of the type of PA recommended to patients with cancer, approximately 51% of oncologists surveyed recommend walking to their patients, and 52.81% of patients said they received that recommendation, probably because it was the simplest to perform and did not require an individualisation plan [38]. According to the roundtable conducted by Campbell et al. (2019) [3]. However, they added that walking was not always enough, especially when it comes to increasing strength, muscle mass, or bone health, where resistance training plays a key role alongside high-impact exercise. Thus, resistance training was the second most recommended type of PA (17.78%) by oncologists, but only 4% of patients have reported the recommendation to perform resistance training, probably because they require specialised advice that oncologists may not have the skills or time to design [32].

Regardless of the activity that patients will be undertaking, the truth is that the main basis for recommending PA starts with a perfectly professional relationship between the healthcare team and those who supervise the training of patients. Schmitz et al. (2019) [39] commented that despite the oncologists should not be in charge of planning specific training, they play a key role in assessing, advising and referring patients to a PA programme, overcoming the gaps between oncologists’ message and what was executed by the patients. In this respect, they proposed a system to encourage PA that involves different health professionals in order to obtain more information for patients’ regular visits and added that more beneficial results could be achieved if PA was performed under supervision.

However, we found that more than 35% of oncologists believe that a PA intervention should be performed by a PA and sports professional, and approximately another 20% stated that it should be conducted by a physiotherapist. This could be due to the barriers encountered by oncologists in recommending PA (lack of time and not knowing how to recommend PA). Nevertheless, these barriers may be considerably reduced in a PA intervention carried out jointly by several health professionals, agreeing with the 10% of oncologists surveyed in this article who believed that this was the most convenient way to organise a PA intervention.

Qualified PA and sports professionals with experience in cancer are apparently thought to be best placed to tailor a PA programme to patients, as they have been found to be preferred by patients. These professionals generate greater patient adherence to the programmes during and after treatment as they were discussed and advised on the plan and know the technique to be followed during training due to supervision [40,41]. Depending on the country, the competencies of health professionals may be different. Therefore, it was important that the person responsible for planning and monitoring patients is legally able to practice in that country, knowing patients’ characteristics and the most suitable PA for them, as well as supporting them with the aim of overcoming barriers since it has been observed that 26.5% of cancer survivors have a lack of motivation as a barrier to exercise and 19.5% do not exercise due to fear of falling [42].

Cancer survivors have been found to significantly increase their frequency and minutes per week of PA, compared to usual care, if they have been recommended some form of PA by their oncologist [43]; similarly, patients who have been recommended PA have walked more days and minutes per week than those who have not received any information on PA [44]. This fact demonstrates the influence that medical doctors can exert on their patients and the responsibility they must take in recommending PA. Overall, knowing that lack of time during patient visits and lack of knowledge about what type of PA is recommended were the main difficulties encountered by oncologists in encouraging PA and the barriers patients encounter to such practice is a good starting point for addressing the inactivity reported by patients with cancer. Perhaps, at the same time, attention should be paid to oncologist–patient communication during clinic visits, as it has been seen that the message is not passing through adequately. Furthermore, it has been observed that the more time is spent addressing the patient’s quality of life, the more visits there were. This may be due to the fact that the first appointments focus the time of the visit on aspects of the treatment itself [45]. On the other hand, they present the idea that a patient-centred conversation with open-ended questions, together with a structured organisation of the visit and active listening of the oncologist, can gather more information. Transferring this to the PA perspective could be a strategy that may help to improve the understanding of the patient’s situation and, thus, achieve a more specific plan, knowing the patient’s limitations or preferences for PA. Different strategies could improve this situation, such as educational interventions to health personnel by physical activity and sports professionals or the improvement of consultation processes and individualisation of exercise prescriptions. Thus, it may lead to greater interest and understanding of the message delivered by the oncologist. Moreover, this survey was administered to any patient regardless of the cancer with which they were diagnosed, making the results more easily generalisable within Spain. In addition, almost 60% of oncologists surveyed indicated that those in charge of PA programmes should be physiotherapists or graduates in PA and sport and, possibly, the lack of this role may be influencing the message transferred and the non-participation of PA by patients.

Finally, we believe that the participation of 93 oncologists from all of Spain was a remarkable and representative sample (6.2% of representativeness) and shows how the health system works with respect to cancer and PA, and these results should be considered for a better projection of training and counselling programmes for patients with cancer. It is true that the sample of patients who participated in the study was relatively small, although it should be noted that they were patients of the oncologists surveyed themselves, which may justify the small sample size.

Previous and ongoing studies have created an important trend that will continue to drive the field of exercise-oncology research for years to come, steadily increasing the evidence in favour of PA prescription in oncology. Additionally, this article provides more insight into the barriers to patients’ not engaging in PA. It seems that one of the reasons was problems in oncologist–patient communication. Thus, this research highlights the need to improve communication protocols in hospitals and oncology practices regarding activities/recommendations for patients with cancer.

In spite of the novelty of the results obtained, this study has certain limitations. It has not been possible to access a very large sample due to the limitations of finding patients linked to an oncologist and both wanting to participate in the study. This fact has probably been a reason for bias in the sample, leaving aside patients or oncologists less committed to physical activity. However, the importance of having collected data directly from the patients of the oncologists evaluated, not from patients not related to those oncologists, must be taken into account. Another limitation may be that oncologists may have answered what they know to be correct or what we wanted to assess without giving an answer to what they actually perform. On the other hand, the fact that the oncologists themselves were the ones who disseminated the questionnaire to their patients may have biased the sample and conditioned the results obtained. Finally, the large difference between men and women oncologists should be mentioned and may have influenced the results obtained.

## 5. Conclusions

Over the last decades, there have been numerous scientific contributions on the importance of PA in patients with cancer. Thus, we have seen that the vast majority of the oncologists participating in this study were in favour of recommending PA for their patients. However, the message does not always seem to pass through to the patients. New strategies are needed to implement a structured and patient-specific plan to improve communication between oncologists and patients, enhancing a referral pathway to support patients and avoid inactivity as much as possible. Moreover, it is believed that a qualified PA and sports professional can help patients overcome barriers to PA and help them build greater adherence to an active lifestyle. Therefore, a collaboration between oncologists and PA professionals can benefit the health care system, providing wisdom through current evidence on oncological PA programmes, analysing, advising, and prescribing a PA plan that is specific and appropriate to the patient, making PA delivered more effectively.

## Figures and Tables

**Table 1 cancers-16-01720-t001:** Patient’s descriptive statistics (n = 149).

Qualitative Variable	Category	n, %
Gender	Female	126, 84.6
	Male	23, 15.4
If you do not perform physical activity, what are the reasons why you do not practice physical activity?	Fear	1, 1.3
	Lack of knowledge of how to perform physical activity properly	9, 12.0
	Following the doctor’s guidelines	4, 5.3
	Disability	8, 10.7
	Lack of resources	2, 2.7
	Lack of time	33, 36.7
	Lack of mood or laziness	8, 12.9
	Tiredness	8, 12.9
	Other	4, 5.5
Has your oncologist recommended That you practice physical activity?	Yes	89, 62.2
	No	54, 37.8
If your answer was affirmative, briefly describe the activities that have been recommended:	Swim	6, 6.7
	Walking	47, 52.8
	Bike	3, 3.4
	Strength training	4, 4.5
	Stretching/mobility activities or exercises	15, 16.9
	Activity in the aquatic environment	1, 1.1
	Others	13, 14.6
Do they reach the minimum of moderate physical activity recommended by the WHO 2020?	Yes	49, 32.7
	No	101, 67.3
**Quantitative Variable**		**Median (IQR)**
Height (cm)		165(9)
Weight (kg)		65(16)
Vigorous PA days/week		1.5(3)
Vigorous PA mins/day		15(60)
Moderate PA days/week		2(4)
Moderate PA mins/day		30(60)
Walking days/week		7(2)
Walking mins/day		60(60)
Sitting hours/day		5(5)
From 0 to 10, what importance would you give to the practice of physical activity in cancer patients?		10(2)

IQR—interquartile range.

**Table 2 cancers-16-01720-t002:** IPAQ survey results are based on the physical activity recommendation status of the patients’ oncologists. (n = 149).

Variable	Has Your Oncologist Recommended that You Practice Physical Activity?		*p*-Value	ES
	Yes (n = 89)	No (n = 53)		
	Median (IQR)	Median (IQR)		
During the past 7 days, how many days did you do vigorous physical activities such as heavy lifting, digging, aerobics, or fast cycling?	1(3)	1.5(3)	0.380	-
How long in total did it usually take you to do vigorous physical activity on one of those days you did it (minutes each day)?	5(60)	20(60)	0.458	-
During the last 7 days, how many days did you do moderate physical activities such as carrying light objects, cycling at a regular pace, or playing tennis doubles? Do not include walks.	1(3)	2(5)	0.138	-
Usually, how much time do you spend on one of those days doing moderate physical activities (minutes each day)?	20(60)	30(60)	0.335	-
During the last 7 days, on how many days did you walk for at least 10 continuous minutes?	7(0)	6.5(3)	0.004	0.442
Usually, how much time did you spend walking on one of those days (minutes each day)?	60(50)	40(60)	0.022	0.410
During the last 7 days, how much time did you sit on one day of the week (hours each day)?	5(5)	5(6)	0.744	-
From 0 to 10, what importance would you give to the practice of physical activity in cancer patients?	10(2)	10(2)	0.737	-

IQR—interquartile range. Oncologists’ results.

**Table 3 cancers-16-01720-t003:** Descriptive statistics on Oncologists, (n = 93).

Qualitative Variable	Category	n, %
Age range	<30 years old	11, 11.8
	30–40 years old	27, 29.0
	41–50 years old	23, 24.7
	51–60 years old	25, 26.9
	>60 years old	7, 7.5
Gender	Female	66, 71.0
	Male	27, 29.0
Do you normally recommend physical activity to your patients?	Yes	90, 96.8
	No	3, 3.2
If your answer was affirmative, briefly describe the recommendations you give them:	Swim	2, 2.2
	Walking	46, 51.1
	Bike	0, 0.0
	Strength training	16, 17.8
	Stretching/mobility activities or exercises	10, 11.1
	Activity in the aquatic environment	0, 0.0
	Other	16, 17.8
If your answer was negative, what are the reasons why you do not recommend it?	Lack of time	2, 50.0
	I would not know what or how to recommend it	1, 25.0
	Others	1, 25.0
Who would you consider most appropriate to carry out an exercise intervention?	Oncologist	16, 17.2
	General practitioner	4, 4.3
	Nurse	8, 8.6
	Physiotherapist	20, 21.5
	Degree in Physical Education	35, 37.6
	Other	10, 10.8
**Quantitative Variable**		**Mean ± SD**
In a normal week, how many days a week do you practice physical activity (walk for more than 20 continuous minutes, run, ride a bike, do strength exercises, swim...)?		3.8 ± 1.6
In a normal week, how many minutes per day of average physical activity? (Walk for more than 20 continuous minutes, run, ride a bike, do strength exercises, swim...)		61.5 ± 35.0
From 0 to 10, how important would it be for cancer patients to practice physical activity?		8.8 ± 1.2

**Table 4 cancers-16-01720-t004:** Oncologists recommend physical activity to their patients according to age groups (n = 93).

Variable	Category	Age Range					*p*-Value
		<30 Years Old (n = 15)	30–40 Years Old (n = 7)	41–50 Years Old (n = 32)	51–60 Years Old (n = 58)	>60 Years Old (n = 37)	
		n, %	n, %	n, %	n, %	n, %	
Do you normally recommend physical activity to your patients?	Yes	10, 11.1	26, 28.9	23, 25.6	24, 26.7	7, 7.8	0.684
	No	1, 33.3	1, 33.3	0, 0.0	1, 33.3	0, 0.0	
If your answer was affirmative, briefly describe the recommendations you give them:	Swim	0, 0.0	0, 0.0	0, 0.0	2, 100.0	0, 0.0	0.016
	Walking *	6, 13.0	17, 37.0	8, 17.4	9, 19.6	6, 13.0	
	Bike	0, 0.0	0, 0.0	0, 0.0	0, 0.0	0, 0.0	
	Strength training *	2, 12.5	5, 31.3	8, 50.0	1, 6.3	0, 0.0	
	Stretching/mobility activities or exercises	0, 0.0	2, 20.0	5, 50.0	3, 30.0	0,0.0	
	Activity in the aquatic environment	0, 0.0	0, 0.0	0, 0.0	0, 0.0	0, 0.0	
	Other *	2, 12.5	2, 12.5	2, 12.5	9, 56.3	1, 6.3	
If your answer was negative, what are the reasons why you do not recommend it?	Lack of time	0, 0.0	1, 50.0	0, 0.0	1, 50.0	0, 0.0	0.287
	I would not know what or how to recommend it	1, 100.0	0, 0.0	0, 0.0	0, 0.0	0, 0.0	
	Others	0, 0.0	0, 0.0	0, 0.0	1, 100.0	0, 0.0	
Who would you consider most appropriate to carry out an exercise intervention?	Oncologist	2, 12.5	6, 37.5	4, 25.0	2, 12.5	2, 12.5	0.243
	General practitioner	2, 50.0	1, 25.0	0, 0.0	1, 25.0	0, 0.0	
	Nurse	0, 0.0	1, 12.5	4, 50.0	3, 37.5	0, 0.0	
	Physiotherapist	2, 10.0	9, 45.0	1, 5.0	5, 25.0	3, 15.0	
	Degree in Physical Education	3, 8.6	9, 25.7	11, 31.4	11, 31.4	1, 2.9	
	Other	2, 20.0	1, 10.0	3, 30.0	3, 30.0	1, 10.0	

* Significant differences (*p* < 0.05) between age ranges.

## Data Availability

The original contributions presented in the study are included in the article; further inquiries can be directed to the corresponding author.

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
