# Peer review of "Do Oncologists Recommend the “Pill” of Physical Activity in Their Practice? Answers from the Oncologist and Patients’ Perspectives"

_cancers, 2024, doi:10.3390/cancers16091720_

Round 1
Reviewer 1 Report
Comments and Suggestions for Authors
The topic addressed by the article is of extreme interest for the promotion of a support intervention through exercise for medical cancer therapies.
We would like to point out the presence of some methodological and interpretation inaccuracies detailed below.
Introduction
The introduction is formulated in a not very fluent, although correct, way. The connections between the different aspects described are sometimes not explicit.
Lines 41 – 45: The first sentences of the introduction are disconnected from each other and it is not clear why the guidelines are indicated for only four pathologies.
Methods
The objective of the research would require a number of subjects, in particular patients, representative of the respective populations.
It seems necessary to report the questionnaires adopted in full.
Lines 107 – 109: I believe that the dissemination of the questionnaire to patients by their doctors should be declared as a critical element because subjects who have received instructions on physical activity from their doctors could be over-expressed.
Lines 118 – 119: It must be specified how the patient questionnaires were sent; in particular it must be specified whether they were sent to the researchers directly by them or through the oncologist.
Results
There is no indication of how many oncologists and potential patients were contacted, thus making the rate of participation in the questionnaires explicit.
Table no. 1 must be arranged to make the correspondence between items and related response values more evident.
Line 161: The number of patients is decidedly limited compared to the oncologists recruited. It is conceivable that many physicians responded to the questionnaire but did not disclose it to their patients. This data confirms the possible underlying flaw in the collection of results in favour of oncologists who are more motivated towards their patients' physical exercise.
Lines 162 – 163: The body mass index would have been a more indicative anthropometric parameter than of weight and height.
Lines 164 – 167: The description of the reasons why patients do not carry out physical activity does not coincide with the values reported in table n.1. In particular, the patients who do not practice physical activity in the table are four (4) while in the description they are indicated as fifty-one (51). Furthermore, almost all the percentages indicated are different.
Line 170: The term "exercises" is not understandable because it does not identify a specific category of exercises or physical activity.
Lines 174 – 178: The IPAQ-SF parameters should also be expressed as a total (MET/1'/week) both in the description and in tables n.1 and n.2.
Line 188: It is appropriate to indicate the total number of oncologists who responded to the questionnaire (it is indicated only in the abstract) and comment on the level of representativeness towards Spanish oncologists. I suggest commenting the great difference between male and female in oncologists.
Lines 188 – 189: In the comment on table n.3, the definition of "middle age" does not correctly interpret the data in the table given that the same concept is used in line 197 to indicate the specific age category between 41 and 50 years.
Lines 193 – 195: You should justify why you analysed oncologists' responses by age group and gender.
Table n.4: In the table header, the distribution of the number of oncologists in the five age groups is different from that described in table n.3
Discussion
Lines 209 – 212: The large difference between the results obtained from the survey and the research with which it was compared raises the suspicion that the methodology for data collection is the bearer of large systematic errors. The statement of the methodological limitations of the research should be formulated more fully.
Lines 218 – 220
It does not seem correct to compare the research results only with the American study (31) considering that, the more recent (2022), Pilotto study (32) also reports similar results.
Lines 240 – 246 and Lines 280 – 285: I suggest comparing your results on barriers to physical activity also with more recent and extensive articles.
Lines 315 – 317: It is not clear why the small number of patients who responded to the questionnaire could be explained by the fact that they are the patients of the doctors who responded. The reason for this statement should be explained.
Author Response
Thanks to the reviewers for their contributions to the manuscript. These contributions have considerably improved the quality of the article. In order to facilitate the reading and observe where the changes in the text have been made, all changes have been marked with control changes. Also, all the answers to reviewers have been answered in blue.
REVIEWER 1
The topic addressed by the article is of extreme interest for the promotion of a support intervention through exercise for medical cancer therapies.
We would like to point out the presence of some methodological and interpretation inaccuracies detailed below.
Thank you so much for your comments.
Introduction
The introduction is formulated in a not very fluent, although correct, way. The connections between the different aspects described are sometimes not explicit.
Dear reviewer, thank you for the comment. Introduction has been rewritten and all the manuscript has been re-read and errors in the connection between concepts have been corrected.
Lines 41 – 45: The first sentences of the introduction are disconnected from each other and it is not clear why the guidelines are indicated for only four pathologies.
Amended.
Methods
The objective of the research would require a number of subjects, in particular patients, representative of the respective populations.
Dear reviewer, we understand your comment, but we could not give a number of potential patients because it was closely related to the number of oncologists who could agree to answer the questionnaire. The patients had to be patients of the participating oncologists.
It seems necessary to report the questionnaires adopted in full.
Dear reviewer, the questionnaire used has been sent to the editor and attached as supplementary material.
Lines 107 – 109: I believe that the dissemination of the questionnaire to patients by their doctors should be declared as a critical element because subjects who have received instructions on physical activity from their doctors could be over-expressed.
Thank you for your contribution. The instruction given to the oncologists participating in the study was to offer to participate in the study to all patients who came to their practice, regardless of the amount of physical activity performed or the observations/advice given to the patients. Despite this, we have added a limitation in the manuscript:
“On the other hand, the fact that the oncologists themselves were the ones who disseminated the questionnaire to their patients may have biased the sample and conditioned the results obtained.”
Lines 118 – 119: It must be specified how the patient questionnaires were sent; in particular it must be specified whether they were sent to the researchers directly by them or through the oncologist.
Dear reviewer, thank you for your feedback. In order to specify how the questionnaire was sent out, the sentence in the manuscript has been modified:
"On the other hand, the second questionnaire contained 15 items and was disseminated through oncologists with a link to google forms for patients to fill in".
Results
There is no indication of how many oncologists and potential patients were contacted, thus making the rate of participation in the questionnaires explicit.
Dear reviewer, the questionnaire was sent to the Spanish Society of Medical Oncology (SEOM) who was the responsible of disseminating it through all the oncologists of Spain. This information has also been added to the methods section.
Table no. 1 must be arranged to make the correspondence between items and related response values more evident.
Dear reviewer, thank you very much for your appreciation. Horizontal lines have been added delimiting each section to better clarify the table and its interpretation.
Line 161: The number of patients is decidedly limited compared to the oncologists recruited. It is conceivable that many physicians responded to the questionnaire but did not disclose it to their patients. This data confirms the possible underlying flaw in the collection of results in favour of oncologists who are more motivated towards their patients' physical exercise.
Thank you for your contribution. Yes, it is a possible hypothesis but we do not have enough information about it. Although we were not able to quantify how many patients did not want to participate in the study and to record the reasons why patients did not want to participate in the study, one of the most common difficulties, which was also present in our study, was the lack of time on the part of the oncologists to explain the study and offer participation to their patients. Some of the oncologists told us that probably because of the poor explanation they could give about the study due to lack of time, some patients might have decided not to participate.
Lines 162 – 163: The body mass index would have been a more indicative anthropometric parameter than of weight and height.
Dear reviewer, you are absolutely right. The reason for not having used the BMI is because the questionnaire, being self-administered, there was a high probability that the patients would not know this data and we would lose information, so it was decided to ask the weight and height in order to make it easier for the patients.
Lines 164 – 167: The description of the reasons why patients do not carry out physical activity does not coincide with the values reported in table n.1. In particular, the patients who do not practice physical activity in the table are four (4) while in the description they are indicated as fifty-one (51). Furthermore, almost all the percentages indicated are different.
Dear reviewer, the results in the table do not agree with the text because they express different things. The table explains the number of patients who answered each question in the questionnaire and in the text, it is described how the percentage of answers is distributed among the most described of those 100% who answered "other reasons".
Line 170: The term "exercises" is not understandable because it does not identify a specific category of exercises or physical activity.
Thank you for the comment. The term “exercises” has been erased from the text.
Lines 174 – 178: The IPAQ-SF parameters should also be expressed as a total (MET/1'/week) both in the description and in tables n.1 and n.2.
Thank you for the comment. The IPAQ questionnaire is not designed to give the results in terms of METS but in days and minutes per week in order to see if they meet the minimum of PA recommended by the WHO.
Line 188: It is appropriate to indicate the total number of oncologists who responded to the questionnaire (it is indicated only in the abstract) and comment on the level of representativeness towards Spanish oncologists. I suggest commenting the great difference between male and female in oncologists.
Thank you for your comment. The total number of oncologists has been added to the text. Also, the level of representativeness towards Spanish oncologists (6,2%) has also been added and the differences between sexes added in the limitations section.
Lines 188 – 189: In the comment on table n.3, the definition of "middle age" does not correctly interpret the data in the table given that the same concept is used in line 197 to indicate the specific age category between 41 and 50 years.
Amended.
Lines 193 – 195: You should justify why you analysed oncologists' responses by age group and gender.
Thank you for you comment. We wanted to analyse the oncologists' responses by age and sex to see if age and sex determined the type of recommendations made. We wanted to know if younger oncologists were more up to date and more likely to recommend PA than older oncologists who may find PA more unusual than traditional drug therapy.
Table n.4: In the table header, the distribution of the number of oncologists in the five age groups is different from that described in table n.3
Dear reviewer, the data in the two tables do not agree because Table 4 refers to the sample of oncologists who did recommend PA.
Discussion
Lines 209 – 212: The large difference between the results obtained from the survey and the research with which it was compared raises the suspicion that the methodology for data collection is the bearer of large systematic errors. The statement of the methodological limitations of the research should be formulated more fully.
Thank you for your comment. Limitations section has been extended.
Lines 218 – 220: It does not seem correct to compare the research results only with the American study (31) considering that, the more recent (2022), Pilotto study (32) also reports similar results.
Dear reviewer, thank you for your appreciation. The information about Pilotto et al. has been added to the text.
Lines 240 – 246 and Lines 280 – 285: I suggest comparing your results on barriers to physical activity also with more recent and extensive articles.
Thank you for this appreciation. We do not want to extend much more the discussion on this aspect because we are about to publish another article based on this questionnaire in which we compare the data obtained in two different countries and in which we go deeper into the aspects you mention and we do not want to be repetitive.
Lines 315 – 317: It is not clear why the small number of patients who responded to the questionnaire could be explained by the fact that they are the patients of the doctors who responded. The reason for this statement should be explained.
Dear reviewer, the number of potential patients to respond to the questionnaire depended directly on the oncologists who decided to participate in the study and responded to the questionnaire. If the oncologist agreed to participate and answered the questionnaire, they could pass the other questionnaire to their patients. The patients must be patients of the oncologists who answered to the questionnaire in order to understand whether the message is getting through correctly and to describe the possible barriers encountered.

Reviewer 2 Report
Comments and Suggestions for Authors
Thank you for letting me read this interesting paper. Examining the disparity between patient reports and advice from oncologists about physical activity is important in cancer care, and the study tackles a relevant problem indeed. A strength of the manuscript is the use of two questionnaires designed for oncologists and patients, respectively. Taken together, the study provides insightful information to the fields of physical exercise and oncology. However, the study needs some improvements regarding the methods and the discussion of the research findings. See below.
1) Introduction. While the purposes of the study are evident, what I feel is missing in this study is a clear definition of why some questions were included in the surveys and which hypotheses the authors wanted to test based on the literature reported in the introduction. This observation points to the importance of aligning survey questions with the study's objectives. The presentation and discussion of research findings could benefit from a more detailed explanation of how each survey question relates to specific hypotheses or existing literature.
2) Regarding 2.1. Study Design, the authors stated that they developed Two online surveys derived from guidelines …… Since several guidelines are mentioned in the introduction and in the reference list, it would be useful for the reader to specify which guidelines were followed in developing the surveys, if any. Another helpful piece of information would be to provide supplementary materials containing the surveys for consultation and potential use in future research.
3) Regarding 2.2. Participants and Procedures, the authors stated that they reached 93 oncologists. How was the sample size determined? What type of sampling method was employed? What was the response rate to the questionnaire sent out? This information is vital to ensure the validity of the results. Typically, a probabilistic sample should be prepared with an appropriate sampling fraction, and the response rate should be high (e.g., 90%). Without these requirements being met, the estimates obtained could be biased, constituting a limitation of the research that needs to be critically discussed.
4) Relatedly, the second questionnaire was distributed to oncologists for patients to complete. In this case, it is important to provide information on sampling within clusters. For example, how many patients were invited by each oncologist? And how many agreed to participate in the research? These details should be included in the text, and if there is a discrepancy in the number of patients per oncologist, the results should be at least weighted accordingly. If the sampling plan was non-probabilistic and/or there was a high refusal rate, the results may contain biases that need to be addressed and critically discussed.
5) In Table 3, Descriptive statistics on Oncologists, I read “In a normal week how many days a week do you practice physical activity?” and “In a normal week, how many minutes per day of average physical activity?”. I have some problems understanding why these questions were included in the survey and how these data were used to understand the gap between the oncologist's prescriptions and the patients' perceptions. Could you clarify?
6) Regarding the discussion, the findings suggest a gap in communication and understanding between oncologists and patients. The paper could discuss potential strategies to improve this, such as educational interventions or enhanced consultation processes. Further discussion on how oncologists can be supported to provide tailored advice to their patients could be beneficial, considering the barriers oncologists faced, such as time constraints and lack of specific training in PA prescription.
Author Response
Thanks to the reviewers for their contributions to the manuscript. These contributions have considerably improved the quality of the article. In order to facilitate the reading and observe where the changes in the text have been made, all changes have been marked with control changes. Also, all the answers to reviewers have been answered in blue.
REVIEWER 2
Thank you for letting me read this interesting paper. Examining the disparity between patient reports and advice from oncologists about physical activity is important in cancer care, and the study tackles a relevant problem indeed. A strength of the manuscript is the use of two questionnaires designed for oncologists and patients, respectively. Taken together, the study provides insightful information to the fields of physical exercise and oncology. However, the study needs some improvements regarding the methods and the discussion of the research findings. See below.
Thank you so much for your comments.
1) Introduction. While the purposes of the study are evident, what I feel is missing in this study is a clear definition of why some questions were included in the surveys and which hypotheses the authors wanted to test based on the literature reported in the introduction. This observation points to the importance of aligning survey questions with the study's objectives. The presentation and discussion of research findings could benefit from a more detailed explanation of how each survey question relates to specific hypotheses or existing literature.
Dear reviewer, the questions asked to both patients and oncologists are short and to the point, aimed at getting their perspective on the amount of PA they receive (patients) or recommend (oncologists). The questionnaires have been sent to the editor as well as attached as supplementary material to the manuscript. On the other hand, we think that the introduction talks about the prescription of PA in cancer and the possible limitation due to several factors and hypothesizes about the possible lack of communication between medical staff and patients.
2) Regarding 2.1. Study Design, the authors stated that they developed Two online surveys derived from guidelines …… Since several guidelines are mentioned in the introduction and in the reference list, it would be useful for the reader to specify which guidelines were followed in developing the surveys, if any. Another helpful piece of information would be to provide supplementary materials containing the surveys for consultation and potential use in future research.
Dear reviewer, thank you for this comment. The guidelines have not been taken specifically from any guidelines, but considering the existing worldwide recommendations (as explained at the European Society of Medicine Oncology ESMO Congress 2024), great importance is given to strength work, cardiovascular work, healthy lifestyle, which has been introduced in the questionnaire in a general way.
3) Regarding 2.2. Participants and Procedures, the authors stated that they reached 93 oncologists. How was the sample size determined? What type of sampling method was employed? What was the response rate to the questionnaire sent out? This information is vital to ensure the validity of the results. Typically, a probabilistic sample should be prepared with an appropriate sampling fraction, and the response rate should be high (e.g., 90%). Without these requirements being met, the estimates obtained could be biased, constituting a limitation of the research that needs to be critically discussed.
Thank you for your comment. These questionnaires were sent via email and WhatsApp containing a link within a brief description of the research to the Spanish Society of Medical Oncology (SEOM) who was the responsible of disseminating it through the oncologists of Spain. We do not have these results you mention. The Spanish health system is very restrictive in terms of data sharing and it is already a great advance to have managed to introduce a questionnaire created at a university into their network of contacts so that they can distribute it. We do not know the total number of oncologists contacted, but we do know that the 93 oncologists are 6.2% of all the oncologists working in Spain (not the sample that was contacted).
Related to this fact we have extended the limitations section.
4) Relatedly, the second questionnaire was distributed to oncologists for patients to complete. In this case, it is important to provide information on sampling within clusters. For example, how many patients were invited by each oncologist? And how many agreed to participate in the research? These details should be included in the text, and if there is a discrepancy in the number of patients per oncologist, the results should be at least weighted accordingly. If the sampling plan was non-probabilistic and/or there was a high refusal rate, the results may contain biases that need to be addressed and critically discussed.
Dear reviewer, as mentioned in the previous question, research groups can not access to data from the Spanish public health system. The participation was voluntary and the obtained sample is a good starting point for the research we are making at the moment.
5) In Table 3, Descriptive statistics on Oncologists, I read “In a normal week how many days a week do you practice physical activity?” and “In a normal week, how many minutes per day of average physical activity?”. I have some problems understanding why these questions were included in the survey and how these data were used to understand the gap between the oncologist's prescriptions and the patients' perceptions. Could you clarify?
Dear reviewer, these questions correspond to the IPAQ questionnaire. This section was introduced in the questionnaire to observe whether the oncologists complied with the minimum weekly PA recommended by the WHO. In future work we are looking at whether or not those oncologists who are physically more active recommend more PA to their patients.
6) Regarding the discussion, the findings suggest a gap in communication and understanding between oncologists and patients. The paper could discuss potential strategies to improve this, such as educational interventions or enhanced consultation processes. Further discussion on how oncologists can be supported to provide tailored advice to their patients could be beneficial, considering the barriers oncologists faced, such as time constraints and lack of specific training in PA prescription.
Thank you so much for the comment. More information about potential strategies has been added to the discussion and conclusion sections.

Reviewer 3 Report
Comments and Suggestions for Authors
The Authors present a very interesting paper: " Do Oncologists recommend the "Pill" of physical activity in their practice? Answers from oncologist and patients' perspectives" and especially very current. PA should be integral part of a cancer related treatment. benefits are recently well documented making strong suggestion to do it. I don't have specific remark to the development of the paper as Introduction, methodology, Data results . Discussion and almost Conclusions should be improved.
In the Discussion that could be shortened the role of the Doctor should be emphasized as well as the type of PA that came out from the questionnaire.
In the Conclusion the take home message should be reported as flow chart:
what from their experience the Authors suggest to do? A sort of memo:
1. who should be in charged to propone the PA
2. what type of exercises they suggest?
3. timing to propone the PA: at diagnosis, during outcome , after treatments?
4. personalized program based on?
Another minor clarification: it is not necessary to replicate this study but extrapolate the important messages and apply them .
Please read also the papers by Lanfranconi et al. about the importance and the role of PA in children and adolescents affected by leukemia and/or cancer.
Comments on the Quality of English LanguageAs to me the English language is good and requires only minor grammar check
Author Response
Thanks to the reviewers for their contributions to the manuscript. These contributions have considerably improved the quality of the article. In order to facilitate the reading and observe where the changes in the text have been made, all changes have been marked with control changes. Also, all the answers to reviewers have been answered in blue.
REVIEWER 3
The Authors present a very interesting paper: " Do Oncologists recommend the "Pill" of physical activity in their practice? Answers from oncologist and patients' perspectives" and especially very current. PA should be integral part of a cancer related treatment. Benefits are recently well documented making strong suggestion to do it. I don't have specific remark to the development of the paper as Introduction, methodology, Data results. Discussion and almost conclusions should be improved.
Thank you so much for your comments.
In the Discussion that could be shortened the role of the Doctor should be emphasized as well as the type of PA that came out from the questionnaire.
In the Conclusion the take home message should be reported as flow chart:
what from their experience the Authors suggest to do? A sort of memo:
1. who should be in charged to propose the PA
2. what type of exercises they suggest?
3. timing to propose the PA: at diagnosis, during outcome, after treatments?
4. personalized program based on?
Dear reviewer, thank you for this comment.
In the projects we have carried out so far with cancer patients and related to PA, we are seeing that the problem is not so much how to train but the fact of training or not training. We are seeing that many patients do not train. That is why the aim of this questionnaire is to find out more about the first line of barriers to PA, whether or not physical activity is prescribed and whether or not this prescription reaches the patient adequately. We do not want to go into what they have to do specifically, but rather whether or not they do or do not do PA as recommended by their oncologist, who is a person in whom they place all their trust at that moment.
The information you mention to add as a take home message we think that it is not directly related to the objective of the study in which we wanted to observe whether oncologists prescribed physical activity in Spain, the variables that influence whether or not to recommend it and to compare these recommendations with the values reported by their patients. The specific information you ask for will shortly be published in another descriptive article.
Anyway, in the conclusions section we have added that “New strategies are needed to implement a structured and patient-specific plan to improve communication between oncologists and patients, enhancing a referral pathway to support patients and avoid inactivity as much as possible”
Another minor clarification: it is not necessary to replicate this study but extrapolate the important messages and apply them.
Dear reviewer, you are right. The ideal is to apply the results obtained in the different health systems and introduce progressive improvements to achieve a program in which the oncologist-patient communication is perfect; and the message of the importance of PA arrives without distortions and in a clear way. All this, giving it the time, it deserves in the medical consultation, information to the patient, family etc.
Please read also the papers by Lanfranconi et al. about the importance and the role of PA in children and adolescents affected by leukemia and/or cancer.
Thank you so much for your recommendation. It is very interesting. We will consider it in future papers with an adolescent sample.
Comments on the Quality of English Language
As to me the English language is good and requires only minor grammar check.
Dear reviewer, the text has been fully revised.

Round 2
Reviewer 2 Report
Comments and Suggestions for Authors
Dear Authors, I have read the revised version of your manuscript. There has been a sufficient improvement in response to the issues I raised in my previous review. I recognize the efforts put forth to address my concerns and I have no further concerns at this stage.